# Application of Polydopamine Functionalized Zinc Oxide for Glucose Biosensor Design

**DOI:** 10.3390/polym13172918

**Published:** 2021-08-30

**Authors:** Viktoriia Fedorenko, Daina Damberga, Karlis Grundsteins, Arunas Ramanavicius, Simonas Ramanavicius, Emerson Coy, Igor Iatsunskyi, Roman Viter

**Affiliations:** 1Institute of Atomic Physics and Spectroscopy, University of Latvia, Jelgavas 3, LV-1004 Riga, Latvia; viktoriia.fedorenko@lu.lv (V.F.); daina.damberga@fizmati.lv (D.D.); karlis.grundsteins@gmail.com (K.G.); arunas.ramanavicius@chf.vu.lt (A.R.); 2Institute of Chemistry, Department of Physical Chemistry, Faculty of Chemistry and Geosciences, Vilnius University, Naugarduko Str. 24, LT-03225 Vilnius, Lithuania; Simonas.Ramanavicius@ftmc.lt; 3NanoBioMedical Centre, Adam Mickiewicz University in Poznan, Wszechnicy Piastowskiej Str. 3, 61-614 Poznan, Poland; coyeme@amu.edu.pl (E.C.); igoyat@amu.edu.pl (I.I.); 4Center for Collective Use of Scientific Equipment, Sumy State University, 31, Sanatornaya Str., 40000 Sumy, Ukraine

**Keywords:** polydopamine (PDA), ZnO-PDA nanocomposite, photo-electrochemical glucose biosensor, ITO modified glass electrode

## Abstract

Zinc oxide (ZnO) nanostructures are widely used in optical sensors and biosensors. Functionalization of these nanostructures with polymers enables optical properties of ZnO to be tailored. Polydopamine (PDA) is a highly biocompatible polymer, which can be used as a versatile coating suitable for application in sensor and biosensor design. In this research, we have grown ZnO-based nanorods on the surface of ITO-modified glass-plated optically transparent electrodes (glass/ITO). Then the deposition of the PDA polymer layer on the surface of ZnO nanorods was performed from an aqueous PDA solution in such a way glass/ITO/ZnO-PDA structure was formed. The ZnO-PDA composite was characterized by SEM, TEM, and FTIR spectroscopy. Then glucose oxidase (GOx) was immobilized using crosslinking by glutaraldehyde on the surface of the ZnO-PDA composite, and glass/ITO/ZnO-PDA/GOx-based biosensing structure was designed. This structure was applied for the photo-electrochemical determination of glucose (Glc) in aqueous solutions. Photo-electrochemical determination of glucose by cyclic voltammetry and amperometry has been performed by glass/ITO/ZnO-PDA/GOx-based biosensor. Here reported modification/functionalization of ZnO nanorods with PDA enhances the photo-electrochemical performance of ZnO nanorods, which is well suited for the design of photo-electrochemical sensors and biosensors.

## 1. Introduction

Photoelectrochemical (PEC) detection is a fast-developing technology [1,2]. Increasing interest in this method is due to its relative simplicity in operation and low cost [3,4,5]. One of the main advantages of photo-electrochemical biosensors compared to conventional electrochemical techniques is lower background noise and higher sensitivity due to the separation of excitation source (light) and analytical signal (photocurrent) [6].

In recent years, considerable attention has been dedicated to the determination of glucose due to its important applications in various fields, such as clinical detection, biological analysis, environmental monitoring, etc. [7,8]. In most glucose sensors, glucose oxidase (GOx) is the crucial component because GOx-catalysed enzymatic reaction is still mostly applied in glucose sensors and is used for the oxidation of glucose into gluconolactone [9]. Recently, GOx-based enzymatic sensors show a rather good selectivity and high sensitivity in glucose determination [10]. However, these sensors still need some improvements [11]. Therefore, investigations related to the development of alternative glucose determination methods photo-electrochemical [12,13,14] are required. Photo-electrochemical biosensor for the determination of glucose and lactose based on TiO_2_ modified with gold nanoparticles and a layer of MnO_2_/g—C_3_N_4_, which was applied for the co-immobilization of glucose oxidase and β-galactosidase has been developed, by this biosensor glucose determination was performed at 0 V potential with a sensitivity of 1.54 μA × mM^−1^ × cm^−2^ within the linear range of 0.004–1.75 mM [15]. An alternative way for glucose determination by realizing photo-electrochemical oxidation of glucose on BiVO_4_-based electrode was proposed. The authors determined a sensitivity of 17.38 μA × mM^−1^ × cm^−2^ towards glucose in the concentration range of 5–35 mM. [16]. Delun Chen et al. proposed a method based on the competitive reaction of ascorbic acid (AA) to enhance the performance of the photo-electrochemical glucose enzyme sensor. It was shown that compared to the detection without AA, the stability of the response current, detection ranges of 1–19 mM, a detection limit of 80 μM, and sensitivity of 2.88 μA × mM^−1^ × cm^−2^ were optimized prominently [17].

Polydopamine (PDA) is a well-known mussel-inspired biocompatible polymer [18]. PDA can be deposited on almost any type of solid substrate, including ceramics, metals, metal oxides, and polymers [19,20]. Recently, there has been much discussion regarding the physical structure of PDA-based layers. Although its physical structure is still a point of discussion, it has proven to be very useful in a large variety of applications, including in electrochemical sensors and biosensors. In sensors, polydopamine is very often used as a binding agent, which improves the characteristics of designed sensors. Polydopamine/graphene/MnO_2_ composite-based electrochemical sensor for the determination of tryptophan in tomato fruit and juice was reported [21]. Another sensor based on a screen-printed carbon electrode, which was covalently modified with self-assembled Au-decorated-PDA nanospheres, was designed. Such a sensor was suitable for a simultaneous determination of ascorbic acid, dopamine, uric acid, and tryptophan [22]. Self-supported nanoporous gold film electrodes, after the functionalization by PDA, demonstrated advantages based on the increased surface area, which were well exploited in the development of sensors suitable for H_2_O_2_ and dopamine determination [23]. In our recent works, we have reported UV light photoinduced processes occurring at the ZnO-PDA interface and evaluated the influence of PDA on some optical properties of formed ZnO-PDA composite. A comprehensive modeling of the processes on the ZnO-PDA interface was shown. The role of the PDA layer on photoluminescence (PL) emission intensity, defect concentrations, and the quantum efficiency in ZnO-PDA-based nanostructures has been represented [24]. Interaction between ZnO-PDA-based structures and poly-l-lysine (PLL) molecules have been studied by photoluminescence spectroscopy [25]. It was determined that changes of ZnO-PDA photoluminescence signal such as the variation of PL-spectral features and increase/decreased of PL-maxima by quenching of PL-emission are observed after PLL adsorption and affected by formed PDA layers, which were formed using different PDA concentrations [25]. Our research illustrates the applicability of PDA coatings for controlling and tailoring some semiconductor surface (e.g., ZnO) properties [24,25]. Thus, due to the specific surface properties of PDA and optical properties of ZnO, the ZnO-PDA nanocomposites have great potential for application in the design of optical sensors.

In this research, we report the application of polydopamine functionalized ZnO nanorods in the design of photo-electrochemical glucose biosensors. For this, we have grown ZnO-based nanorods on the surface of ITO modified glass plated electrode and then formed ZnO-based nanorods were coated by PDA. Then glucose oxidase (GOx) was immobilized on the surface of the ZnO-PDA composite. Photo-electrochemical determination of glucose has been performed by glass/ITO/ZnO-PDA/GOx-based electrode using potentiodynamic and potentiostatic methods.

## 2. Experimental

### 2.1. Materials and Instruments

Zinc acetate dehydrate, hexamethylenetetramine, 2-propanol (IPA), ethanolamine, sodium sulfate, zinc nitrate hexahydrate, and phosphate-buffered saline (tablet) were purchased from Sigma Aldrich (Darmstadt, Germany), dopamine hydrochloride 99%, and tris(hydroxymethyl)aminomethane 99% (A18494) were purchased from Alfa Aesar (Poland), and were used without any further purification. Glucose oxidase was purified from Aspergillus niger, ~360 U/mg protein (approx. 280 U/mg material), and purchased from CarlRoth (Germany). Glucose anhydrous pure p.a. (C_6_H_12_O_6_–180.16 g/mol) was purchased from Chempur (Poland).

The ITO glass substrates were cleaned by successive sonication with deionized water and isopropyl alcohol for 10 min, with proper drying prior to final use. Oxygen plasma treatment for 15 min was performed in order to eliminate organic traces.

Structural properties of the ZnO-PDA nanostructures were investigated by scanning electron microscope (SEM) (Zeiss Evo HD15 SEM from Zeiss Ltd. (Jena, Germany)), and high-resolution transmission electron microscope (HR-TEM) (JEOL ARM 200F, Tokyo, Japan) (200 kV) with energy-dispersive X-ray spectroscopy (EDX) and electron energy-loss spectroscopy (EELS) detector and FTIR spectroscopy was performed using an FTIR-ATR spectrophotometer ‘Frontier’ from Perkin Elmer (Waltham, MA, USA). The HI98129 Combo tester from Hanna Instruments was used to measure the pH of Tris and PBS buffers.

### 2.2. Synthesis of ZnO Nanorods and the Deposition of Polydopamine

ZnO nanorods (ZnO-NRs) were formed by the hydrothermal method (Scheme 1A). Briefly: ZnO seed-based layer was prepared on ITO glass by drop-casting of 20 µL zinc acetate Zn(CH_3_COO)_2_ of 5 mM in methanol solution and annealed at 350 °C for 1 h. Then, the substrates with ZnO seed layers were incubated for 4 h in 50 mM of zinc nitrate and 50 mM of hexamethylenetetramine-containing solution in water at 95 °C. Finally, the samples were washed with deionized water and dried at room temperature. As prepared, ITO glass substrates modified by ZnO-NRs were immersed into a Tris buffer (10 mM, pH 8.5, 50 mL) containing a dopamine concentration of 0.5 mg mL^−1^ at room temperature for 1 h. After that, the samples were removed and rinsed with water, which was purified using the Milli-Q system and then dried at room temperature. Then, glucose oxidase (GOx) was deposited on the surface of ZnO-PDA nanostructures from 50 mM phosphate buffer (PBS) solution, pH 7.2, containing 10 mg/mL of GOx. It was incubated at room temperature for 40 min and then cross-linked using a 0.1% aqueous solution of glutaraldehyde. Samples exposed to the chamber after 3 cycles of washing in Milli-Q water glass/ITO/ZnO-PDA/GOx-based electrodes were used for photo-electrochemical measurements by using potentiodynamic and potentiostatic methods.

### 2.3. Photoelectrochemical Detection of Glucose

A homemade plastic cuvette, equipped with front quartz glass, was used for electrochemical measurements (Scheme 1B). Three electrode configuration was used for all electrochemical measurements (working electrode, Pt counter electrode, and reference Ag/AgCl electrode).

The glass/ITO/ZnO-PDA/GOx-based electrodes were excited by UV LED (365 nm, 4 mW, 15 nm full width at half maximum (FWHM)). Potential cycling was performed in potential range of −500–800 mV at potential sweep rate of 50 mV, and cyclic voltammograms (CVs) were registered. Chronoamperometry-based evaluation at constant potential also was performed ‘at dark’ and at illumination by UV light. The hexacyanoferrate-based electrochemical redox probe was used for all photo-electrochemical experiments. All measurements were performed at room temperature.

The determination of glucose was performed according to the following scheme: glucose was added into an electrochemical cell, which was continuously stirred for 60 s to distribute glucose homogeneously in the cell; after homogeneous distribution of glucose, cyclic voltammetry and chronoamperometry based measurements were performed. Glucose concentration in electrochemical cells varied from 0 to 20 mM.

## 3. Results and Discussion

SEM and TEM images of ZnO-PDA nanorods deposited on glass/ITO surfaces are presented in Figure 1. The average dimensions of nanorods were 60 nm in diameter and 800 nm in length (Figure 1A); this observation is in line with previously reported data [26]. PDA formed a layer over ZnO nanorods and in between them (grey patterns in Figure 1B). TEM measurements (Figure 1C) illustrate the formation of a 7 nm thick PDA layer over ZnO.

Fourier transform infrared spectroscopy (FTIR) was used to characterize the immobilization of glucose oxidase on the ZnO-PDA surface (Figure 1D). As prepared, ZnO-PDA nanocomposite structures were characterized by peaks, located at 1288 cm^−1^, 1492 cm^−1^, 1607 cm^−1^, and 3362 cm^−1^ that corresponded to C–O, C=N or/and C=C, C=O and –OH or/and N–H vibrational modes, respectively (Figure 1D). These modes are in agreement with previous data for PDA observed by Hongyong Luo et al. [27]. In the mentioned research, FTIR spectra were chosen to investigate the binding interactions between polydopamine sphere (PDS) and silver nanoparticles (AgNPs). The PDS characteristic peaks (1292 cm^−1^, 1512 cm^−1^, 1627 cm^−1^, and 3379 cm^−1^) are correlated to peaks observed for ZnO/PDA in our study. The difference of peak position (4–20 cm^−1^), observed in noted and our study, could be due to different protocols of polydopamine synthesis. The ZnO/PDA contains active quinone groups that can react with the amino groups of the GOx through Michael addition and/or Schiff base reaction and then resulted in immobilization of GOx onto the surfaces of the ZnO/PDA. A new peak located at 1063 cm^−1^ appeared after the immobilization of glucose oxidase (Figure 1D). The registered new peak corresponds to C–O bending vibrational mode [28,29]. The immobilization of glucose oxidase resulted in the decrease of peak intensity at 1607 cm^−1^. It is important to note that the immobilization of glucose oxidase on the ZnO-PDA surface leads to shifting the FTIR peak positions (1492 cm^−1^, 1607 cm^−1^, and 3362 cm^−1^) of 12–20 cm^−1^ to higher values of wavenumbers. The peak shifting could be observed due to the reaction between active quinone groups of ZnO/PDA and amino groups of GOx. Thus, the peak shifting could prove the successful GOx immobilization on ZnO/PDA.

The characterization of the samples before and after forming of GOx layer was performed by using cyclic voltammetry (Figure 2A,B). It was found that GOx deposition resulted in significant decrease of the current. Adding of the glucose probe resulted in enhancement of the current through the system.

Chronoamperometry was applied for the determination of glass/ITO/ZnO-PDA/GOx electrode-based sensor response towards glucose. The response of the sensor was calculated as a difference between current registered under illumination by UV light and the current value measured at ‘dark’ conditions (Equation (1)).
(1)$$I_{S}=I_{UV}-I_{dark}$$
where *I_uv_* and *I_dark_* are current values, measured at UV excitation and at ‘dark’ conditions.

The sensor response S was calculated as relative change of the sensor signal according to the following equation:
(2)$$S=1-\frac{I_{S}\left(C\right)}{I_{S}\left(0\right)}$$

The response of the sensor towards different glucose concentrations is represented in Figure 3.

The response of glass/ITO/ZnO-PDA/GOx electrode-based sensor increased in a glucose concentration range of 0.015–0.12 mM, and the saturation of analytical signal was determined at glucose concentrations exceeding 0.120 mM. The limit of detection (LOD) of this glucose biosensor was determined as 0.0062 mM.

The photoinduced processes in metal oxide nanocomposites are still under discussion [13,30,31,32]. Therefore, the interaction mechanisms and principles of detection in such systems still require additional analysis. In metal oxide photo-electrochemical glucose sensors, the excitation light results in a significant increase of the photocurrent [13,30,31,32]. At ZnO, photogenerated charge participates in redox reactions catalyzed by glucose oxidase. The interaction between the sensor surface and target molecules involves conductivity electrons; therefore, the photocurrent after the treatment of glass/ITO/ZnO-PDA/GOx electrode-based sensor with glucose has lower values [13,30,31,32]. PL quenching in ZnO-PDA nanostructures points to that the formation of organic layer over the ZnO induces additional charge separation and, therefore, enhancement of surface catalytic activity [24,26]. Therefore, the photocurrent changes could be more pronounced when compared to the ‘dark’ current changes [26]. Similar to the methodology, which is reported in [26], changes between photocurrent and ‘dark’ current were determined. Due to the catalytic action of glucose oxidase in the presence of glucose, the ‘dark’ current has slightly increased. Therefore, the correlation between ‘dark’ and photocurrent provides the possibility to measure glucose concentration more accurately.

Recent results related to the development of glucose photo-electrochemical sensors based on composite nanomaterials show sensitivity towards glucose in the range of 0.01–2 mM [13,30,31,32]. The developed sensor showed good sensitivity towards glucose in the range of 0.015–0.12 mM. It is known that physiological glucose concentration levels are in the range of 1.5–20 mM [13,30,31,32]. Therefore, the sensitivity of here reported glucose biosensor over 50 times exceeds analytical requirements for such sensors, which enables to apply of these sensors for the determination of glucose in very diluted samples and/or the modification of here proposed structure by diffusional membranes, that can change the sensitivity of developed biosensors. Hence, the developed biosensor shows sensitivity and response time are suitable for the application of glass/ITO/ZnO-PDA/GOx based electrodes in the determination of glucose in biological samples. 

## 4. Conclusions

In summary, we have observed a significant decrease of current due to the GOx layer formed on the ZnO-PDA structure. Catalytic oxidation of glucose by glucose oxidase resulted in a concentration-dependent photo-electrochemical response of glass/ITO/ZnO-PDA/GOx-based electrode towards glucose. Chronoamperometric signals were measured at UV-illumination and in the ‘dark’, and the difference of measured amperometric signals was interpreted as an analytical signal suitable for the determination of glucose concentration in the sample. Using this method, fast response and reliable sensor response were registered in the glucose concentration range of 0.0062–0.120 mM.

Potential application of glass/ITO/ZnO-PDA structures modified by some other oxidases and/or other redox enzymes can be predicted in photo-electrochemical sensors for the determination of compounds oxidized by these enzymes.

## Data Availability

Not applicable.

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
