# Peer review of "Application of Polydopamine Functionalized Zinc Oxide for Glucose Biosensor Design"

_polymers, 2021, doi:10.3390/polym13172918_

Round 1

Reviewer 1 Report

The authors described ZnO-based nanorod sensor for glucose sensing, the manuscript is well prepared in general. the authors have addressed the reviewers' comments point-to-point. In figure 4, replicates of experiments should be done and show error bars.

Author Response

Dear reviewer

Reviewer 2 Report

Nel manoscritto intitolato “Application of Polydopamine Functionalized Zinc Oxide for Glucose Biosensor Design”, gli autori mostrano la sintesi di un biosensore fotoelettrochimico del glucosio costituito da nanobarre di ZnO cresciute sulla superficie vetro/ITO rivestita con uno strato di polimero PDA e su cui il glucosio ossidasi è stata immobilizzata. Questo composito è stato caratterizzato da spettroscopia infrarossa SEM, TEM e trasformata di Fourier.

Raccomando di pubblicare questo articolo su Polymers, dopo importanti revisioni. Di seguito i commenti dettagliati:

  • In the abstract, the authors write that they have characterized the biosensor by TEM, SEM, and Raman spectroscopy. Actually, it was used Fourier transform infrared spectroscopy was used to characterize the immobilization of glucose oxidase on the ZnO-PDA surface. Therefore, I ask the authors to correct this error and to describe FTIR in the Materials and Instruments section. Furthermore, a better technique in being able to confirm the deposition of both glucose oxidase and the PDA layer is X-ray photoelectron spectroscopy (XPS). I ask the authors to perform XPS measurements to verify what happens in the electronic system during the various synthesis steps. How does the electronic structure of nanorods grown on the ITO and after functionalization with PDA and glucose oxidase change? Furthermore, should the authors also carry out XPS measurements of the biosensor after the electrochemical measurements to check if any chemical changes have occurred that would compromise its reuse?
  • lines 108-109, Please the author should specify the acronyms SEM and HR-TEM.
  • line 113, it is better to replace the word “Figure 1” with “Scheme 1” because the authors are describing the synthetic process and the setup for the electrochemical measurements. Furthermore, they should include in the letter a) of the schematic diagram how the seed layer formation was done.
  • In the Materials and Instruments section, the authors did not describe the instrument used to measure the pH of Tris buffer containing dopamine and the phosphate buffer (PBS) if they were freshly prepared. Alternatively, if the PBS has been purchased, they should always specify its characteristics in the Materials and Instruments section.
  • How many moles of zinc acetate (PM = ?) are there deposited on the ITO? Moreover, the authors are requested to clarify the importance of using hexamethylenetetramine with zinc nitrate and their mole ratio.
  • lines 132 and 149, authors should use the lowercase “a” for the word “After”.
  • Is the aqueous solution of glutaraldehyde 0.1% of by weight or by mole with respect to?
  • My curiosity is, what do the authors mean when they talk about incubation during the biosensor synthesis?
  • The authors show no investigation of the surface chemical composition of ZnO functionalized with both PDA and glucose oxidase. XPS is a suitable technique for displaying this information which should be compared with the EDX analysis. Indeed, the authors describe this technique but do not show any EDX images on the prepared systems.
  • Authors should be careful in saying that the PDA layer surrounding ZnO nanorods, observed in TEM measurements, is 7 nm thick because it is not homogeneous and uniform. Furthermore, they could highlight this by distinguishing the crystalline planes of the ZnO nanorods with the amorphous layer of the PDA molecules surrounding them.
  • Per favore, gli autori dovrebbero anche mostrare lo spettro FTIR di ZnO-ITO nella Figura 2D. Inoltre, gli autori dovrebbero mostrare la reazione chimica che si verifica tra lo strato PDA e la glucosio ossidasi per essere più chiari nella descrizione dei segnali FTIR. Sono coinvolte la formazione di legami covalenti o semplici interazioni deboli? Gli spettri FTIR mostrano spostamenti di 12-20 cm-1 a valori di numero d'onda più elevati dei segnali dopo l'immobilizzazione della glucosio ossidasi sulla superficie ZnO-PDA. A cosa sono dovuti questi enormi cambiamenti? Meglio chiarire alcune considerazioni.
  • Meglio sostituire tutte le (4) “Fig.” con "Figura" nel testo.
  • riga 166, a cosa si riferisce la Fig. 3 mentre si discutono gli spettri FTIR?
  • Meglio scrivere l'inizio delle conclusioni perché la frase alla riga 223 non è chiara.
  • L'inglese deve essere un po' migliorato, poiché ci sono alcune congiunzioni abusate e difetti tecnici da correggere nel manoscritto.

Author Response

Dear reviewer.

Round 2

Reviewer 2 Report

I thank the authors for answering my questions.

Author Response

Dear Reviewer.

Thank you for your work.

We will make changes on English language in updated version of the manuscript. 

This manuscript is a resubmission of an earlier submission. The following is a list of the peer review reports and author responses from that submission.

Round 1

Reviewer 1 Report

This project developed a glucose biosensor by using glucose oxidase immobilized on the polydopamine coated zinc oxide nanorods. It displayed high sensitivity.

Since this report is a research article, not a review as labeled in the submission, there are many details missing in this manuscript.  

  1. the average size/diameter of nanorods need to be measured and reported.
  2. resource of glucose oxidase used in this research and its activity need to be given. 
  3. a titration of glucose oxidase for immobilization and the sensitivity need to be performed and analyzed. 
  4. the condition of the glucose sample used for the analysis of the response of the sensor. 
  5. do protein and other components affect the response of this sensor.

Reviewer 2 Report

The authors of the manuscript entitled “Application of polydopamine functionalized Zinc oxide for glucose biosensor design” did not bring any improvement in comparison with a several works already done on glucose biosensors. Furthermore, polydopamine is widely employed as binding agent.

  • The linear range of the developed sensor is 0.015 -0.12 MM of glucose which is much lower than the physiological concentration level 1.5 – 20 mM.
  • The stability and lifetime of the proposed senor were not investigated.
  • The selectivity of the sensor was not investigated as well.

In the light of the above reasons, I will not recommend publication of this manuscript in this Journal “Polymers”.